# An Improved Prediction Model for Ovarian Cancer Using Urinary Biomarkers and a Novel Validation Strategy

**DOI:** 10.3390/ijms20194938

**Published:** 2019-10-05

**Authors:** Shin-Wha Lee, Ha-Young Lee, Hyo Joo Bang, Hye-Jeong Song, Sek Won Kong, Yong-Man Kim

**Affiliations:** 1Department of Obstetrics & Gynecology, University of Ulsan, ASAN Medical Center, Seoul 05505, Korea; swhlee@amc.seoul.kr; 2ASAN Institute for Life Science, ASAN Medical Center, Seoul 05505, Korea; leehayoung@gmail.com; 3Ahngook Pharmaceutical Co., Ltd., Seoul 07445, Korea; jwnme@ahn-gook.com; 4Bio-IT Research Center, Hallym University, Chuncheon, Gangwon-do 24252, Korea; hjsong@hallym.ac.kr; 5Computational Health Informatics Program, Boston Children’s Hospital, Boston, MA 02115, USA; sekwon.kong@childrens.harvard.edu

**Keywords:** ovarian cancer, prediction model, urinary biomarker

## Abstract

This study was designed to analyze urinary proteins associated with ovarian cancer (OC) and investigate the potential urinary biomarker panel to predict malignancy in women with pelvic masses. We analyzed 23 biomarkers in urine samples obtained from 295 patients with pelvic masses scheduled for surgery. The concentration of urinary biomarkers was quantitatively assessed by the xMAP bead-based multiplexed immunoassay. To identify the performance of each biomarker in predicting cancer over benign tumors, we used a repeated leave-group-out cross-validation strategy. The prediction models using multimarkers were evaluated to develop a urinary ovarian cancer panel. After the exclusion of 12 borderline tumors, the urinary concentration of 17 biomarkers exhibited significant differences between 158 OCs and 125 benign tumors. Human epididymis protein 4 (HE4), vascular cell adhesion molecule (VCAM), and transthyretin (TTR) were the top three biomarkers representing a higher concentration in OC. HE4 demonstrated the highest performance in all samples with OC (mean area under the receiver operating characteristic curve (AUC) 0.822, 95% CI: 0.772–0.869), whereas TTR showed the highest efficacy in early-stage OC (AUC 0.789, 95% CI: 0.714–0.856). Overall, HE4 was the most informative biomarker, followed by creatinine, carcinoembryonic antigen (CEA), neural cell adhesion molecule (NCAM), and TTR using the least absolute shrinkage and selection operator (LASSO) regression models. A multimarker panel consisting of HE4, creatinine, CEA, and TTR presented the best performance with 93.7% sensitivity (SN) at 70.6% specificity (SP) to predict OC over the benign tumor. This panel performed well regardless of disease status and demonstrated an improved performance by including menopausal status. In conclusion, the urinary biomarker panel with HE4, creatinine, CEA, and TTR provided promising efficacy in predicting OC over benign tumors in women with pelvic masses. It was also a non-invasive and easily available diagnostic tool.

## 1. Introduction

Ovarian cancer (OC) is the seventh most common malignancy among women worldwide, accounting for 3.6% of cancers in women [1]. It is the most lethal of all gynecological malignancies, with more than 70% of women presenting with the advanced-stage disease [2], and the primary treatment for advanced-stage disease involves both cytoreductive surgery and chemotherapy. Despite aggressive treatment, long-term survival has not significantly improved in the past 30 years, with the five-year survival rate remaining between 30% and 45% [3]. Pelvic masses are often a diagnostic challenge for physicians, as discriminating malignant from benign masses as early intervention is crucial for prognosis. Patients with advanced-stage OC have a 20–40% five-year survival rate, in contrast to a >90% five-year survival rate for patients identified with and treated for stage-I OC [4]. The management of OC with complete cytoreductive surgery by a gynecologic oncologist enables a favorable prognosis [5]; therefore, the decision of an appropriate referral is important. Unfortunately, it has been reported that 65% of patients with malignant adnexal masses are inappropriately referred to gynecologic oncologists in the United States of America [6].

Cancer antigen 125 (CA-125) is the most widely used biomarker in OC; however, it lacks adequate diagnostic sensitivity (SN) and specificity (SP) to be used reliably as a screening test. Recently, in a randomized controlled trial, annual multimodal screening with serum CA-125 and transvaginal ultrasound sonography as a second-line test revealed limited efficacy when prevalent cases were excluded [7]. The Risk of Malignancy Index (RMI) was developed to assess the malignancy risk in women with adnexal masses, which incorporates a woman’s CA-125 level, ultrasound morphology score, and menopausal status [8]. Two algorithms have been developed for the same reason. The Risk of Malignancy Algorithm (ROMA) is based on serum levels of human epididymis protein 4 (HE4) and CA-125 according to menopausal status, and OVA1 includes five serum biomarkers: CA-125, β-2 microglobulin, transferrin, transthyretin (TTR), and apolipoprotein [9,10,11]. Various studies have been published evaluating the effectiveness of RMI, ROMA, and OVA1 tests, as well as other strategies to distinguish the risk of malignancy in the setting of a pelvic mass. These are not screening tests for women in certain age groups, but could be used as decision-supporting tools for surgical intervention. However, to date, definitive conclusions have not been drawn regarding these tests due to a lack of prospective multi-institutional trials and cost–benefit analyses. Furthermore, the low compliance of patients for transvaginal ultrasound sonography and blood tests is another unresolved issue.

For these reasons, discovering biomarkers in urine can provide a non-invasive method for detecting OC, and enable the frequent testing of women who belong to high-risk groups. The protein profile in the urine is also less complex than that in the blood, and thus the measurement of urinary proteins may also enhance overall clinical performance. Several biomarkers have been identified in the urine of women with OC including eosinophil-derived neurotoxin, a fragment of osteopontin, mesothelin, and Bcl-2 [12,13,14]. The level of HE4 in the urine has been reported to have a relationship with primary platinum resistance that is not evident in serum levels [15]. However, there is a limit to the clinical applications of urinary proteins for patients with OC, and published studies have only been presented for a single biomarker with a simple statistical association. Here, we report an extended analysis of 23 protein biomarkers in the urine associated with OC and a panel of urinary biomarkers that demonstrate high performance in distinguishing OC from benign ovarian tumors in patients with several types of pelvic mass.

## 2. Results

### 2.1. Urinary Protein Biomarkers in Patients with Ovarian Cancer

Figure 1 shows the modeling process used to find the optimum biomarkers andthe best combination. Twelve patients who were diagnosed with borderline tumors after surgery were excluded from the analysis. After exclusion, 283 urine samples were analyzed. The mean age of patients was different in the benign group and the cancer group (mean ± standard deviation: 40.6 ± 12.2 vs 53.8 ± 10.4, respectively). Among the 158 patients with OC, 70.3% exhibited papillary serous adenocarcinoma, 69.6% had advanced-stage cancer (International Federation of Gynecology and Obstetrics (FIGO) stage III and IV), and 58.2% exhibited postmenopausal status (Table 1). The urinary concentration of 17 biomarkers showed significant differences between OC and benign tumors (corrected *p* < 0.05). The concentrations of HE4, vascular cell adhesion molecule (VCAM), TTR, macrophage migration inhibitory factor (MIF), leptin, C-reactive protein (CRP), platelet-derived growth factor (PDGF)-AA, Cyfra21-1, neural cell adhesion molecule (NCAM), prolactin, myeloperoxidase (MPO), Mesomark, CA19-9, and apolipoprotein A1 (ApoAI) were significantly higher in the urine of patients with OC than in those with benign tumors, whereas the concentrations of carcinoembryonic antigen (CEA), creatinine and plasminogen activator inhibitor-1 (PAI-1) were significantly lower in patients with OC than in those with benign tumors. HE4, VCAM, and TTR were the top three altered biomarkers in this analysis and, among them, HE4 exhibited a significant difference with a maximum *p*-value of 1.26 × 10-24. CA-125 is a traditional serum tumor biomarker in OC, but it was not significant in our urine data (Appendix A).

Next, we tested whether the urine concentration of each protein marker could classify OC from benign tumors. We used a bootstrap resampling method to calculate the mean area under the receiver operating characteristic (AUC) from 2000 randomly chosen subsets while maintaining the OC-to-benign tumor ratios, and 95% confidence intervals (CIs) were estimated using DeLong’s method. Table 2 shows the performance of each single biomarker in urine samples for predicting cancer over benign masses. HE4 exhibited the greatest AUC values in total samples of OC followed by VCAM, leptin, TTR, and prolactin. Interestingly, TTR showed the highest performance in early-stage cancers, whereas HE4 demonstrated the highest value in advanced-stage cancers.

### 2.2. Urinary Multimarker Panel Analysis in Patients with Ovarian Cancer

To build and compare the performance of multimarker prediction models, we used the Least Absolute Shrinkage and Selection Operator (LASSO) regression model to prioritize each marker. Using all biomarkers in the current study, we built LASSO models to predict OC over benign tumors with cross-validations as implemented in the R library glmnet. For each iteration of cross-validation, the regularization parameter lambda was determined. The magnitude of lambda for each marker was summarized from 2000 iterations of cross-validation to rank the importance of each biomarker. HE4 is the most influential biomarker in the differential diagnosis between OC and benign tumors, followed by creatinine, CEA, NCAM, and TTR (Figure 2). Interestingly, HE4 demonstrated not only the maximum t-score but also the greatest coefficient in the logistic regression model. However, the subsequent t-score biomarker VCAM revealed a small coefficient. Several biomarkers (e.g., NCAM) had a wide variation in the distribution coefficients. Notably, this coefficient result was closely related to the ROC curve of the next panel analysis.

When the bootstrap resampling method was applied to the multivariate analysis, the top-performing two-biomarker combinations identified each included HE4 and creatinine. This combination outperformed the best individual biomarkers in terms of SN, SP, and AUC values. The top-performing three-biomarker panel included combinations of HE4, creatinine, and CEA. This panel provided notably elevated AUC values in comparison to the two-biomarker combinations and individual biomarkers, and it was modestly increased after multimarker combinations. Our final model included HE4, creatinine, CEA, and TTR; this panel provided 93.7% SN and a 78.7% positive predictive value (PPV) as well as 90.6% negative predictive value (NPV) for all-stage OC at 70.6% SP. When adding the menopausal status as an independent variable to the selected panel, it was confirmed that the AUC value increased, even if at very weak levels (Figure 3 and Table 3).

Each of the biomarker panels identified in this study was further evaluated for the disease status in patients with early-stage cancer and advanced-stage cancer. The four-biomarker panel with HE4, creatinine, CEA, and TTR was found to be the most useful for the classification of cancer compared with that of benign tumors regardless of disease status. In the early-stage group, in particular, the performance of multimarker models used to predict cancer over benign tumors improved remarkably compared with that in the advanced-stage group. When adding menopausal status as an independent variable to the selected panel, the AUC value increased regardless of the FIGO stage (AUC: 0.932 vs 0.933 in the early-stage group, 0.946 vs 0.951 in the advanced-stage group) (Figure 4).

### 2.3. Clinical Characteristics of Difficult Samples to Predict

We checked whether the samples that were not correctly predicted by our multimarker panel shared any clinical characteristics. Among 15 patients with OC representing a false-negative in our result, seven patients (46.7%) exhibited early-stage disease, including two patients with mucinous intraepithelial cancer, one with malignant granulosa cell tumor, and one with dysgerminoma. Among eight patients with advanced-stage disease, six patients revealed low-grade serous adenocarcinoma, and the remaining two patients had clear cell carcinoma and endometrioid adenocarcinoma. Our multimarker panel did not fail to identify the high-grade serous OC that accounts for most OCs, even in early-stage disease. The clinical characteristics of patients representing a false-negative in this study were summarized as nonepithelial OCs and early-stage non-serous OCs. In contrast, looking at two patients with a false-positive, we could confirm necrosis in the surgical specimens caused by the torsion of teratoma or ischemia of a hemorrhagic cyst. Therefore, rare ovarian malignancies and necrotic ischemic benign lesions should be considered if the results of biomarkers do not match the clinical judgment.

## 3. Discussion

A multimarker panel appears to serve an essential role in the differential diagnosis of OC in women who have pelvic masses [16,17,18,19,20]. In particular, many investigators are currently performing biomarker research using urine, as urine samples can be acquired truly non-invasively (as opposed to blood samples) [15,21,22,23,24]. The goal of this study was to determine a panel of biomarkers that exhibit high SN and SP in order to distinguish OC from benign ovarian tumors in patients with the heterogeneous characteristics of tumors at different growth stages using bead-based xMAP multiplexing technology and truly non-invasive urine samples. Firstly, in agreement with published evidence [15,22], HE4 in urine was the most important biomarker for ovarian neoplasm. However, although HE4 was a strong biomarker for predicting the overall stages of OC (particularly early stage), TTR provided better performance than HE4. This is a good reason to develop a multimarker panel for the early detection and differential diagnosis of OC.

After performing a multimarker multiplexed immunoassay, we investigated (in line with reported evidence [16]) that four-biomarker panels seem to offer superior performance compared with two- and three-biomarker panels. Through repeating the 10-fold cross-validation 2000 times, we identified a combination of the top four urinary protein biomarkers (i.e., HE4, creatinine, CEA, and TTR) that provided 94% SN, 79% PPV, and 91% NPV for all-stage OC with 71% SP. SP in the final analysis was relatively low, which is why the objective of this study was a biomarker for differential diagnosis and not for screening. For the screening test, a high SN should be achieved for distinguishing women with early-stage OC from healthy individuals with at least 98% SP. Due to the low prevalence of OC (accounting for 3.6% of cancers in women worldwide) [1], the required minimum SP of the serum test should be 98% when combined with transvaginal sonography (TVS) as a second-line test [25,26], and accurate assessment of such high SP require at least 1000 healthy participants. However, the present results are comparable to other differential diagnostic techniques using blood samples: (1) the urinary protein panel (our results) offered 93% SN at 75% SP, (2) the RMI showed 85–96% SN at 75% SP [27,28], (3) the ROMA provided 94% SN at 75% SP [27,28,29], and (4) the OVA1 test revealed 92–97% SN at a low SP of approximately 54% [29,30] (Table 4).

Although the performance of the differential diagnosis of OC was similar to that of other diagnostic panels approved by the US Food and Drug Administration (FDA), our urinary panel had distinct advantages as a diagnostic test. This was because urine is an excellent alternative liquid biopsy for biomarker research, as it is available in larger quantities compared to blood and through a less invasive process, allowing for repeated tests. The identification of urinary biomarkers could provide a more convenient and accurate test for diagnosing malignant tumors. The urinary proteome is a direct product of renal filtration and consists of soluble, low molecular weight peptides that are highly appropriate for proteomic analyses that may represent disease-specific cleavages. Renal filtration also results in a less complex matrix compared to the serum and contains fewer factors known to interfere with biomarker assays, which can be an advantage in a proteomic analysis [12,13,24].

There were some limitations to this study, including the small sample size and 23 selected protein biomarkers. In the current study, the biggest limitation was the absence of an independent validation set. However, we improved the significance of our data by performing a resampling method and multivariate statistical analyses, which were applied to an in vitro diagnostic multivariate index assay (IVDMIA) of proteomic biomarkers. The most important method was to create random sampling and maintain independence for the logistic regression analysis by a bootstrap resampling method. Bootstrapping is the practice of estimating properties of an estimator (e.g., variance) by measuring properties when sampling from an approximate distribution, and can refer to any test that relies on random sampling [31,32,33]. Among algorithms for comparing the areas under two or more correlated ROC curves, DeLong’s algorithm is the most widely used due to its simplicity of implementation in practice [17,34]. In this way, we were able to maximize the independence of the hold-out testing set of samples [17]. The second method was the LASSO regression. Variable selection in the regression analysis is a key procedure when we have a large collection of possible covariates for the efficient prediction of a response variable. LASSO not only helps to improve prediction accuracy when dealing with multicollinear data, but also has several outstanding properties such as interpretability and numerical stability [35]. The last limitation was the certainty of creatinine as a biomarker. Urinary creatinine concentration is changeable based on age, race/ethnicity, body mass index, and time of day. These factors were not controlled in this study. However, other biomarkers from urine change following creatinine concentration. Therefore, when using multiple biomarkers, the interaction among biomarkers is more important rather than factors associated with the creatinine level.

## 4. Materials and Methods

### 4.1. Patient Populations and Urine Sample Collections

This study was an observational cohort study with a population of 295 patients diagnosed with ovarian tumors and scheduled for surgery. We obtained their urine samples prospectively before the pathological diagnosis, and excluded 12 cases with borderline tumors after enrollment. Other inclusion criteria were that patients be over the age of 18 years, agree to urine collection before surgery, and provide written informed consent. Exclusion criteria were patients who underwent surgical or open biopsy for any reason within the past 28 days, patients with clinically significant medical problems, and patients who were diagnosed with other cancers within the past 5 years. Urine samples were obtained before the operation as the first-morning urine after more than 8 hours of fasting, and were frozen at −70 or −80 °C within 1 hour of collection until testing. Each urine sample was annotated with information regarding age, menopausal status, histology, and the International Federation of Gynecology and Obstetrics (FIGO) stage (in epithelial ovarian cancer) (Table 1). A collection of samples at the ASAN Medical Center was performed according to strict protocols approved by the Institutional Review Board of the ASAN Medical Center (AMC IRB 2012-0067, approved on 10 February 2012).

### 4.2. Multiplexed Urinary Biomarker Analysis

The xMAP bead-based technology (Luminex Corp., Austin, TX) facilitates the multiplexed analysis of multiple analytes in a single sample. The 23 bead-based xMAP immunoassays for the differential diagnosis of OC used in this study are listed in Table 5. This list of biomarkers was compiled based on a literature review of current proteins of interest within all fields of circulating biomarkers related to OC research, based on suitable immunoassay availability. The concentration of the biomarkers in urine was calculated by a microbead-based antibody multiplexed immunoassay using Luminex, and the multiplexed immunoassay kit including the cancer biomarkers specifically reacting to OC was used. Bead-based immunoassays targeting 23 protein biomarkers were developed according to strict quality control standards by the University of Pittsburgh Cancer Institute (UPCI) Luminex Core Facility [36] and were performed according to manufacturers’ protocols as previously described [16]. As an exception, all urine samples were tested undiluted. Complete biomarker testing was performed immediately upon thawing without testing protein concentration or any other pretest manipulation. The total protein content of each urine sample was measured using the Bio-Plex suspension array system (Bio-Rad Life Science Research, Hercules, CA). Biomarker expression levels were expressed as the median fluorescent intensities generated by analyzing 100 microbeads for each analyte in each sample. The concentrations of the analytes were then quantitated from the median fluorescence intensities using standard curves generated by the Bio-Rad five-parameter curve fitting to the series of known concentrations of each analyte [37].

### 4.3. Statistical Analysis

The development of statistical models for distinguishing patients with OC from benign tumors was continued until one panel and one model of combining the candidate biomarkers in the panel was selected. The likelihood of OC for each sample was compared to the post-operative confirmatory diagnosis, to evaluate the accuracy of the model. For each biomarker, we used a subsampling strategy to rank markers according to the mean area under the receiver operating characteristic curve (AUC) from 2000 random subsamplings for 80% of the dataset, while maintaining the OC and benign tumor ratio throughout. Using top-performing markers, we constructed multimarker prediction models. To determine the best biomarker combination, 10-fold cross-validation was repeated 2000 times and the top biomarker combinations were selected using the average AUC. By repeating the cross-validation 2000 times, the deviation between the total set of combinations and the chosen subset could be decreased. The combination selected consisted of 2–4 biomarkers out of 23, and the score threshold for the logistic regression was determined using the F-score. Panels were also evaluated based on the sensitivity (SN) at predetermined specificity (SP) levels between 70% and 95%. These SP criteria were selected from reference data on diagnostic products (ROMA and OVA1) that the US FDA had approved for the discrimination of OC from patients with a pelvic mass. Due to the concern that discrepancies in menopausal status may contribute to biomarker panel performance, the final algorithm was applied to the biomarker results in a separate analysis that included menopausal status as an independent variable.

We used a nested cross-validation technique to minimize over-fitting to the dataset. Briefly, 80% of samples were randomly selected to create a training set (i.e., outer loop). In a training set, we used repeated cross-validations to maximize the performance of multimarker models (i.e., inner loop), and the overall performance of a panel was evaluated with the remaining 20% of samples. The OC-to-benign tumor ratio was maintained throughout the resampling procedure [38]. AUC was used as a performance measure, and we used DeLong’s method to determine the statistical difference between two AUC values [17,31,32,33,34]. Figure 1 shows the modeling process to find the optimum biomarkers and classify. Comparisons between different sample sources were performed using a one-way analysis of variance (ANOVA) with Tukey’s multiple comparison tests. An initial minimum level of significance of *p* < 0.05 was utilized. All statistical analyses were performed using the statistical language R.

## 5. Conclusions

In conclusion, the performance of the HE4, creatinine, CEA, and TTR panel for the differential diagnosis of OC in women with pelvic masses appears promising, with high levels of SN and acceptable SP. Our results suggest that urinary biomarkers provide diagnostic properties exceeding those reported for serum biomarkers by applying not only accuracy but also non-invasiveness. Assuming a patient has a pelvic mass and does not need immediate surgery (but is at high risk of OC, including a history of breast cancer, familial history of breast/ovarian cancer, or BRCA mutation), she would require continuous check-up. Moreover, a non-invasive accurate test is needed for such a patient. Further evaluation of this panel and other candidate urinary biomarkers (particularly in early-stage OC) would, through prospective studies, expand the clinical utility of urinary multimarker panels.

## Figures and Tables

**Figure 1 ijms-20-04938-f001:**
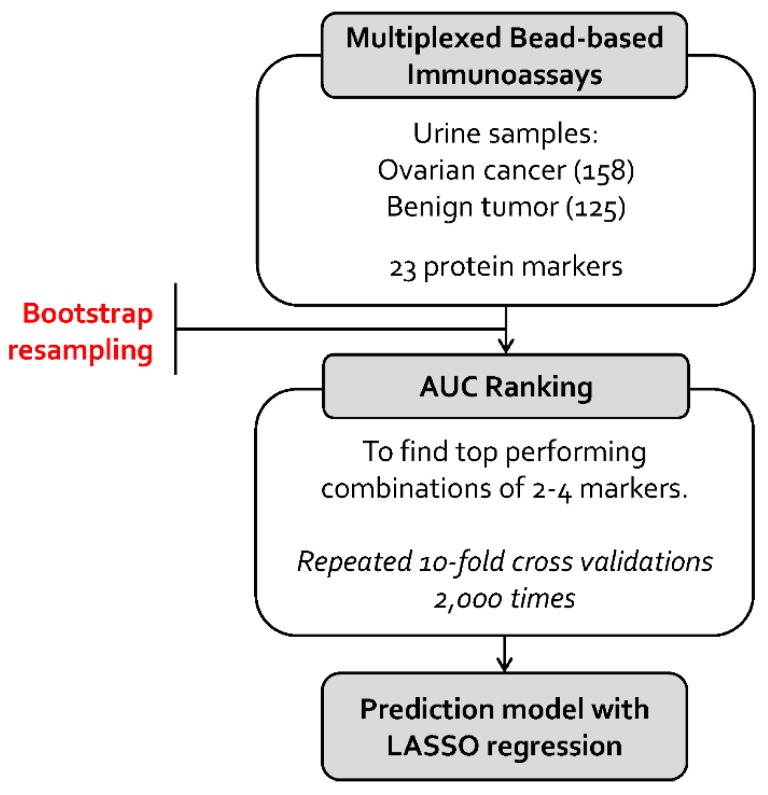
Process of the differential diagnosis modeling between ovarian cancer (OC) and benign tumor in this study. AUC, area under the receiver operating characteristic curve; LASSO, Least Absolute Shrinkage and Selection Operator.

**Figure 2 ijms-20-04938-f002:**
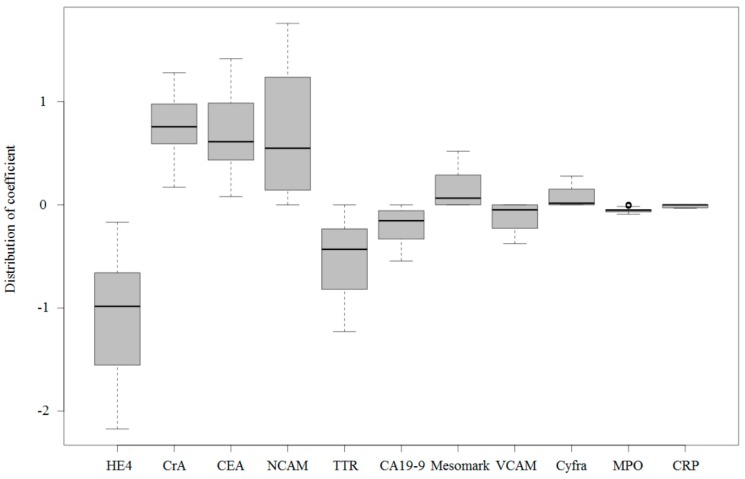
Distribution of the coefficient by the LASSO regression method. The y-axis is the distribution coefficient of the logistic regression model analyzed by the LASSO regression method. There are multiple values because the method was repeated 10-fold, with 2000 iterations of cross-validation. The mean value is in descending order from the left. As the absolute value is large, it will have a large coefficient from the model. In this sense, when the standard deviation of the measured value is changed to 1, it affects the results of the entire model. HE4 is the greatest contributor, followed by creatinine, CEA, NCAM, and TTR. There is large variation in the case of NCAM, and VCAM does not have a large contribution.

**Figure 3 ijms-20-04938-f003:**
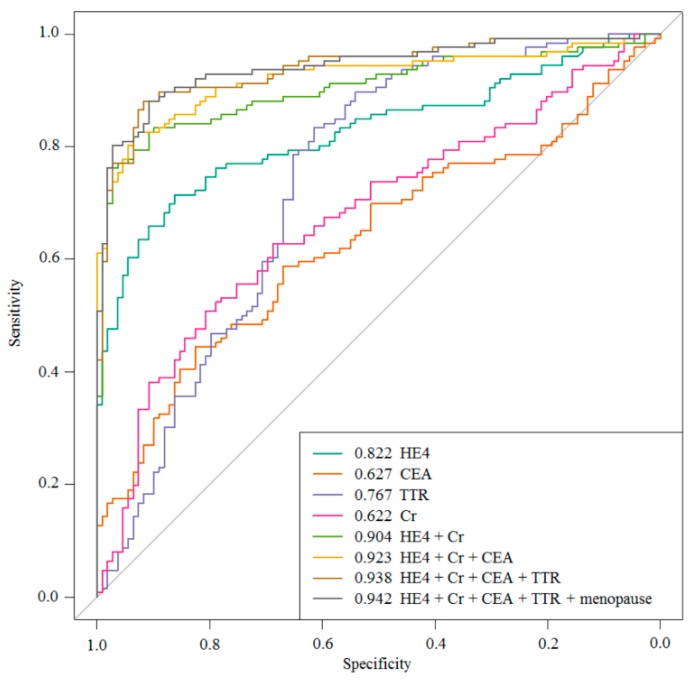
The ROC curve of the multimarker panels (i.e., HE4, creatinine, CEA, and TTR), and the results with the addition of menopausal status.

**Figure 4 ijms-20-04938-f004:**
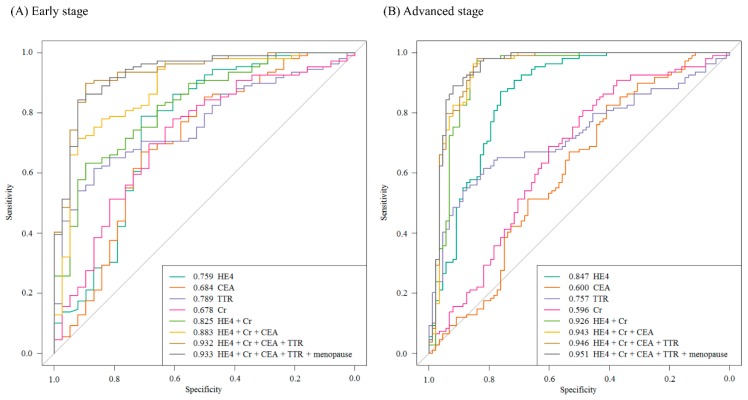
The ROC curve of multimarker panels (i.e., HE4, creatinine, CEA, and TTR) according to the FIGO stage, and the results from the addition of menopausal status.

**Table 1 ijms-20-04938-t001:** Clinical and demographic characteristics of the study patients.

	Number of Totals	Premenopausal	Postmenopausal
Total	295	175	119
Age (years), median and range	48 (20–82)		
Benign tumor	125 (42.4%)	99	25
Endometriosis	44	41	3
Teratoma	30 *	27	2
Mucinous cystadenoma	16	13	3
Serous cystadenoma	9	5	4
Inflammation ^1^	5	1	4
Others ^2^	21	12	9
Borderline tumor	12 (4.1%)	10	2
Malignant tumor	158 (53.5%)	66	92
Serous adenocarcinoma	111	40	71
Mucinous adenocarcinoma	12	6	6
Endometrioid adenocarcinoma	15	11	4
Clear cell carcinoma	12	4	8
Other EOC ^3^	3	1	2
Granulosa cell tumor	2	2	-
Dysgerminoma	1	1	-
Other non-EOC ^4^	2	1	1
FIGO stage of malignancy			
I	36	23	13
II	12	5	7
III	91	32	59
IV	19	6	13

EOC, epithelial ovarian cancer; FIGO, the International Federation of Gynecology and Obstetrics. * The menopausal status of one patient was not stated; ^1^ three tubo-ovarian abscesses/two chronic granulomatous inflammations; ^2^ eight paratubal cysts, six hemorrhagic cysts, five fibromas, and two leiomyomas; ^3^ two transitional cell carcinomas and one mixed cell type (endometrioid and serous adenocarcinoma); ^4^ two mixed germ cell tumors.

**Table 2 ijms-20-04938-t002:** The performance of each single biomarker in the urine samples to predict cancer over benign tumors.

Markers	All Samples		Stage I and II		Stage III and IV	
	AUC	(95% CIs)	AUC	(95% CIs)	AUC	(95% CIs)
HE4	0.822	(0.772–0.869)	0.759	(0.659–0.852)	0.847	(0.791–0.898)
VCAM	0.776	(0.717–0.829)	0.744	(0.650–0.830)	0.788	(0.725–0.847)
Leptin	0.771	(0.701–0.837)	0.779	(0.657–0.889)	0.772	(0.689–0.848)
TTR	0.767	(0.706–0.824)	0.789	(0.714–0.856)	0.757	(0.693–0.819)
Prolactin	0.713	(0.624–0.793)	0.732	(0.573–0.870)	0.701	(0.600–0.794)
CRP	0.710	(0.644–0.772)	0.644	(0.541–0.743)	0.734	(0.662–0.800)
PDGF-AA	0.697	(0.632–0.758)	0.734	(0.644–0.820)	0.677	(0.607–0.747)
NCAM	0.678	(0.613–0.741)	0.672	(0.576–0.761)	0.678	(0.609–0.742)
Mesomark	0.670	(0.578–0.756)	0.648	(0.520–0.769)	0.680	(0.583–0.769)
MPO	0.668	(0.598–0.737)	0.640	(0.554–0.723)	0.684	(0.610–0.756)
Cyfra21-1	0.660	(0.591–0.726)	0.728	(0.630–0.821)	0.628	(0.549–0.701)
CEA	0.627	(0.558–0.692)	0.684	(0.583–0.778)	0.600	(0.524–0.676)
Creatinine	0.622	(0.554–0.687)	0.678	(0.585–0.770)	0.596	(0.518–0.668)
CA19-9	0.598	(0.529–0.666)	0.578	(0.470–0.678)	0.604	(0.528–0.677)
IL6	0.576	(0.450–0.701)	0.490	(0.323–0.657)	0.599	(0.459–0.730)
MIF	0.572	(0.500–0.642)	0.640	(0.531–0.743)	0.537	(0.456–0.618)
ApoAI	0.557	(0.485–0.623)	0.517	(0.422–0.612)	0.569	(0.493–0.644)
ApoCIII	0.524	(0.448–0.600)	0.580	(0.466–0.687)	0.562	(0.479–0.644)
PAI-1	0.523	(0.446–0.598)	0.514	(0.417–0.616)	0.533	(0.451–0.611)
CA125	0.523	(0.453–0.591)	0.486	(0.370–0.597)	0.533	(0.456–0.611)
OPN	0.521	(0.45–0.591)	0.540	(0.443–0.635)	0.514	(0.435–0.594)
IL8	0.488	(0.416–0.563)	0.541	(0.449–0.633)	0.531	(0.453–0.607)
CA15-3	0.486	(0.417–0.552)	0.540	(0.439–0.638)	0.536	(0.460–0.613)

AUC, area under the receiver operating characteristic curve; CIs, confidence intervals. AUC for each biomarker to predict cancer over benign tumors. The 95% confidence intervals were calculated using a bootstrap resampling method. We repeated resampling 2000 times with a replacement in proportion to the cancer-to-benign ratio of all samples. HE4, human epididymis protein 4; VCAM, vascular cell adhesion molecule; TTR, transthyretin; CEA, carcinoembryonic antigen; NCAM, neural cell adhesion molecule; CA-125, cancer antigen 125; CRP, C-reactive protein; PDGF, platelet-derived growth factor; MPO, myeloperoxidase; IL, interleukin; MIF, macrophage migration inhibitory factor; ApoAI, apolipoprotein A1; ApoCIII, apolipoprotein C3; PAI-1, plasminogen activator inhibitor-1; OPN, osteopontin.

**Table 3 ijms-20-04938-t003:** The performance of multimarker models to predict cancer over benign tissues between early- and advanced-stage cancer (this analysis was performed using 109 benign and 131 cancer samples).

Markers	All Samples		Stage I and II		Stage III and IV	
	AUC	(95% CIs)	AUC	(95% CIs)	AUC	(95% CIs)
HE4	0.822	(0.772–0.869)	0.759	(0.659–0.852)	0.847	(0.791–0.898)
CEA	0.627	(0.558–0.692)	0.684	(0.583–0.778)	0.600	(0.524–0.676)
TTR	0.767	(0.706–0.824)	0.789	(0.714–0.856)	0.757	(0.693–0.819)
Creatinine (Cr)	0.622	(0.554–0.687)	0.678	(0.585–0.770)	0.596	(0.518–0.668)
HE4+Cr	0.904	(0.854–0.938)	0.825	(0.740–0.910)	0.926	(0.888–0.972)
HE4+Cr+CEA	0.923	(0.878–0.954)	0.883	(0.772–0.934)	0.943	(0.907–0.982)
HE4+Cr+CEA+TTR	0.938	(0.900–0.964)	0.932	(0.844–0.970)	0.946	(0.911–0.983)

AUC, area under the receiver operating characteristic curve; CIs, confidence intervals. AUC for each biomarker to predict cancer over benign tumors. The 95% confidence intervals were calculated using DeLong’s method. We repeated resampling 2000 times with replacement in proportion to the cancer to benign ratio of all samples.

**Table 4 ijms-20-04938-t004:** Specificity and sensitivity results of various screening strategies in the setting of a pelvic mass.

Algorithm or Assay		N	SN (%)	SP (%)	PPV	NPV	AUC
*Serum Multimarker Panels*							
Moore, 2010 [27]	RMI	457	84.6	75.0	-	-	0.870
	ROMA	457	94.3	75.0	-	-	0.953
Karlsen, 2012 [28]	RMI	1218	96.0	75.0	-	-	0.958
	ROMA	1218	94.8	75.0	-	-	0.954
Bristow, 2013 [30]	OVA1	494	92.4	53.5	31.3	96.8	-
Grenache, 2015 [29]	ROMA	146	83.9	83.5	57.8	95.1	-
	OVA1	146	96.8	54.8	36.6	98.4	-
*Urinary Biomarker*							
Hellstrom, 2010 [22]	HE4	135 ^1^	88.6	94.4			
Liao, 2015 [15]	HE4	279 ^2^	52.2	95.0			
*Urinary Multimarker Panel in this Study*							
HE4+Cr+CEA+TTR		283	81.0 92.9	95.3 75.2	95.3 81.3	81.3 90.1	0.938

SN, sensitivity; SP, specificity; PPV, positive predictive value; NPV, negative predictive value; AUC, area under the receiver operating characteristic curve; ^1^ including 79 patients with ovarian cancer; ^2^ including 92 patients with ovarian cancer; RMI, Risk of Malignancy Index; ROMA, Risk of Malignancy Algorithm.

**Table 5 ijms-20-04938-t005:** Complete list of multiplexed biomarkers.

Inflammatory Mediators	IL-6 ^1^, IL-8 ^1^, MPO ^1^, MIF ^1^, OPN ^1^
Tumor-associated antigens	CA19-9 ^1^, CA15-3 ^1^, CA-125 ^1^, HE4 ^1^, CEA ^1^
Adhesion molecules	VCAM^1^, NCAM ^1^
Adipokines	Leptin ^1^
Apolipoproteins	ApoAI ^1^, ApoCIII ^1^
Apoptotic proteins	Cyfra21-1 ^1^
Growth/angiogenic factors	PDGF-AA ^1^
Carrier proteins	TTR ^2^
Proteases/inhibitors	PAI-1 ^1^
Hormones	Prolactin ^1^
Others	CRP^1^, Mesomark ^3^, Creatinine ^4^

^1^ Merck Millipore corp., Darmstadt, Germany; ^2^ Abnova corp., Taipei, Taiwan; ^3^ Fujirebio Diagnostics, Inc., Göteborg, Sweden; ^4^ Abcam plc., Cambridge, UK.

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
