# Peer review of "An Improved Prediction Model for Ovarian Cancer Using Urinary Biomarkers and a Novel Validation Strategy"

_ijms, 2019, doi:10.3390/ijms20194938_

Round 1
Reviewer 1 Report
In the present study, Yong-Man Kim et al. investigated the level of several biomarkers in the urine of patients with benign tumor and ovarian cancer cells. The authors determined if the potential urinary biomarker panel can predict malignancy. They found that urinary biomarker panel with HE4, creatinine, CEA, and TTR can predict advanced ovarian cancer over benign tumors in women. This is an interesting study with clinical relevance. As the authors mentioned, the biggest limitation of this study is the absence of an independent validation set.
Comments:
The authors did not mention, but they should test if there is any confounding or interaction in their study. If possible, measure the level of identified biomarkers in the urine of patients after surgery can give much more information.Author Response
Reply to Reviewer 1
ID: IJMS-581541
TITLE: An improved prediction model for ovarian cancer using urinary biomarkers and novel validation strategy
I really appreciate your comments. I agree and concern in your point of view. In order to exclude any confounding or interaction we need to perform an independent validation set, as mentioned. Currently this study seems to be significant to suggest a novel way of testing set for biomarkers. In the future, validation set, I will try to measure the level of identified biomarkers in the urine before and after surgery. Besides, I revised many things in manuscript. Please read precisely.
Thank you so much.
Reviewer 2 Report
Comments:
This is a well-written manuscript evaluating 23 biomarkers for the diagnosis of ovarian cancer. Authors evaluated urine biomarkers with a prospective enrollment of patients with adnexal mass. The ovarian cancer topic and increasing interest on non-invasive diagnostic biomarkers will have an impact on IJMS readers. However, there are major concerns with the manuscript, including study population and methodology which needs to be addressed before this manuscript becomes suitable for publication.
Comments:
In abstract section, I recommend introducing abbreviations including HE4, VCAM, and TTR as they might be foreign markers to readers. Authors stated in Abstract that “HE4 demonstrated the highest performance in all samples with OC, whereas 6TTR showed the highest efficacy in early-stage OC.” It is not clear for readers if performance or efficacy means highest sensitivity. I recommend more clear definition in abstract as it is the first reflection of the manuscript. Abstract should clearly state the number of cases and controls. Authors stated that 295 patients included in the study but in the abstract there is no mention of 158 ovarian cancer patients that were compared to 125 controls. Additionally there were 12 borderline cases that were excluded so 295 is not reflective of samples that were analyzed. The number of cases vs controls is imperative for readers who will be reading the abstract prior to full manuscript. Introduction is overall well written. However, it is not clear why non-epithelial ovarian cancer patients were included into this analysis. Authors cited ROMA, OVA1 index at the introduction which are used for the detection of epithelial ovarian cancer (EOC). They also evaluated HE4, CA125 in this setting where they are known to be markers for EOC. However, there is a small group of non-epithelial ovarian cancer patients (n=5) included, which is not statistically significant. It will be worthwhile to see how these markers perform solely on EOC cases. Same limitation accounts for CA19-9 and mucinous cancers. There are only 12 mucinous cases which might be the limitation of insignificance of CA19-9. Author’s performed a prospective recruitment. Additionally, if 12 borderline patients were excluded from the final analysis. Based on the author’s description, cases and controls were determined following pathology analysis. If a case – control study was performed, did authors match cases with controls to eliminate confounding factors? Authors should clearly define the mean/median age, menopausal status of cases vs controls and if there was any statistically significant difference. Table 1 presents the number of premenopausal and postmenopausal patients. Supplementary table 1 also presents similar data. Authors should consider removing one of them to reduce replicates. Additionally, table 1 states that there were 175 premenopausal patients vs 119 postmenopausal. It is not clear again how many of these were cases vs controls. Based on the Supp Table 1, 92 ovarian cancer patients were postmenopausal which would yield only 27 postmenopausal controls, which is proportionally not matched with cancer patients. Many tumor markers including Ca125 have different clinical interpretation based on menopausal status. Ca125 was not statistically different between cases and controls and it is not clear whether menopause status was a confounder as this marker will increase in benign premenopausal cases. Additionally, authors included menopausal status into their regression model while the ratio was not adjusted between cases and controls. Authors need to include the histopathology of benign cases. It may be beneficial to seen how these markers perform between EOC cases vs epithelial benign masses. Additionally, did authors exclude patients with a history of inflammatory, renal disease, previous cancer? Many markers presented in the manuscript are known to be cancer / inflammatory markers (CRP, IL-6, IL-8, CEA, CA 19-9..) Authors stated in methodology section that urine samples were obtained before the operation, after fasting, and were frozen at −70°C or −80°C within 1 hour of collection until testing. Did they apply any centrifugation prior to storage? This is imperative to clearly define SOP of the sample collection. Did any of the ovarian cancer patients receive any neo-adjuvant chemotherapy? If they did, this is a significant confounding factor, where these patients need to be excluded. As 19 patients had FIGO IV stage disease, for optimal cytoreduction, I expect there is a chance that they received such a treatment. Authors used LASSO technique with cross validation for regression models. Authors performed cross-validation to minimize over fitting which is very helpful. However, the choice of the ratio (80% training vs 20% test set) needs to be discussed. There is a statement in results section “In contrast, looking at two patients with false-positive, we could confirm the necrosis in the surgical specimens caused by torsion of teratoma or ischemia of hemorrhagic cyst. Therefore, rare ovarian malignancies and necrotic ischemic benign lesions should be considered if the results of biomarkers do not match the clinical judgment.” This needs to be moved to discussion section. Supplementary Table 2 presents mean and standard deviation for each marker. In the methodology section, authors need to explain the distribution of their data. Are all these markers normally distributed? If a biomarker not normally distributed, I recommend using median with interquartile range, also state their statistical methodology based on the distribution. In discussion section, authors compared their regression model with previously reported diagnostic models. This comparison along with sensitivity and specificity values are welcome, however, all the cited detection models are for EOC cases. Authors included non EOC cases in their model, which should be revised before comparison.The section on 'difficult samples to predict' is welcome, but this does not eliminate the mis-comparision of all OC cases with cited EOC markers. Urinary creatinine was a part of biomarker model, it should be discussed that urinary creatinine concentration is due to change based on age, race/ethnicity, body mass index, and time of day (Barr et al. Environmental Health Perspectives Vol. 113, No. 2) In conclusion section, authors mention for potential use of these markers in BRCA mutation patients. As BRCA mutation patients are under risk for other cancers, with the modest specificity, this biomarker model needs to be tested in different cancer types prior validating ovarian cancer detection for BRCA positive patients.Author Response
Reply to Reviewer 2
ID: IJMS-581541
TITLE: An improved prediction model for ovarian cancer using urinary biomarkers and novel validation strategy
I really appreciate your comments. I agreed some correction marked in manuscript. And, in order to improve contents, I revised many things in manuscript. Please read precisely and see the attachment
Thank you so much.

Reviewer 3 Report
ID IJMS-581541
" An improved prediction model for ovarian cancer using urinary biomarkers and novel validation strategy" to be published in International Journal of Molecular Sciences.
General remarks
The authors tried to analyze urinary proteins associated with ovarian cancer and investigate the potential urinary biomarker panel to predict malignancy in women with pelvic masses. They analyzed 23 biomarkers in urines samples obtained from 295 patients with pelvic masses scheduled for surgery. The concentration of urinary biomarkers was quantitatively assessed by the xMAPTM bead-based multiplexed immunoassay. To identify the performance of each biomarker to predict cancer over benign, they used a repeated leave-group out cross validation strategy. The prediction models using multi-makers were evaluated to develop a urinary ovarian cancer panel. The urinary concentration of 17 biomarkers exhibited significant differences between ovarian cancer and benign tumor. HE4, VCAM, and TTR were the top three-biomarkers representing a higher concentration in ovarian cancer. HE4 demonstrated the highest performance in all samples with ovarian cancer, whereas TTR showed the highest efficacy in early-stage ovarian cancer. Overall, HE4 was the most informative biomarker, followed by creatinine, CEA, NCAM, and TTR using the least absolute shrinkage and selection operator (LASSO) regression models. A multi-marker panel consisted with HE4, creatinine, CEA, and TTR presented the best performance with 93.7% sensitivity at 70.6% specificity to predict ovarian cancer over the benign tumor. This panel performed well regardless of the disease status and demonstrated an improved performance by including the menopausal status. Authors concluded that the urinary biomarker panel with HE4, creatinine, CEA, and TTR provides promising efficacy to predict OC over benign tumors in women with pelvic masses. They stated, that it is also a non-invasive and easily available diagnostic tool.
The topic of the article is very interesting and it could serve as an important basis for further research.
There are only some minor corrections needed before the article is suitable for publication, marked in the attached manuscript.
Author Response
Reply to Reviewer 3
ID: IJMS-581541
TITLE: An improved prediction model for ovarian cancer using urinary biomarkers and novel validation strategy
I really appreciate your comments. I agreed some correction marked in manuscript. And, in order to improve contents, I revised many things in manuscript. Please read precisely.
Thank you so much.
Round 2
Reviewer 2 Report
Author's described their study not being a case control study. I recommend stating the design of this study at methodology section.
If this study was a prospective cohort study, authors should explain why borderline tumor cases were excluded from analysis?
I appreciate Table 1 which includes the details of the histopathologies. I recommend presenting the mean age of benign controls vs cases? This will be pertinent to readers.
What was the reason of not centrifugating urine prior to testing?
Under the benign controls, there are 5 patients with diagnosis of inflammation. This will be important to state the clinical diagnosis of inflammation cases, like Tuboovarian abscesses.
On the other hand, authors included creatinine in their regression models. As it was previously stated in the comments, creatinine is due for change due to age and cancer cases vs controls are not age matched. They comment that they controlled this during multi biomarker analysis however, logistic regression models including creatinine is most likely to be effected by this factor.
Author Response
Reply to Reviewer
ID: IJMS-581541
TITLE: An improved prediction model for ovarian cancer using urinary biomarkers and novel validation strategy
I really appreciate your comments.
I revised manuscript as you mentioned and added authors’ opinion regarding centrifuge and creatinine as a biomarker. And my manuscript has been edited by MDPI English editing.
Thank you so much, again.
|
Comments |
Reply |
1 |
stating the design of this study at methodology section |
Revised sentence in methodology section: The study was the observational cohort study with population comprised 295 patients diagnosed with an ovarian tumor and scheduled for surgery. |
2 |
authors should explain why borderline tumor cases were excluded from analysis |
Added sentence in methodology section: We obtained their urine samples prospectively before the pathological diagnosis, so we excluded 12 cases with borderline tumor for the analysis after enrollment. |
3 |
presenting the mean age of benign controls vs cases |
Added sentence in result section: The mean age of patients was different in benign group and cancer group (mean ± standard deviation; 40.6 ± 12.2 vs. 53.8 ± 10.4, respectively). |
4 |
the reason of not centrifugating urine prior to testing |
We freeze samples immediately after obtaining without centrifugating. We did centrifuge prior to testing after defreezing urine samples. |
5 |
clinical diagnosis of inflammation cases |
Added explanation below Table 1: 3 tubo-ovarian abscess/2 chronic granulomatous inflammation |
6 |
logistic regression models including creatinine is most likely to be effected by creatinine |
In this point, I’m not sure. Actually, we don’t know which of age and cancer is more important to the level of creatinine. We did not correct the level of each proteins according to creatinine because of multi-biomarker study, and final statistical analysis revealed that creatinine worked as one of biomarkers to discriminate cancer patients. |
Round 3
Reviewer 2 Report
I recommend stating couple points as the limitations of this study prior to publication.
1) Authors should mention that they were not able to match age between cases and controls due to design of this study
2) Secondly, as authors replied to creatinine comment, readers should know this as a limitation of this study. They should state that due to age discrepancy between two groups, it is not clear if the significance of creatinine is affected by age or cancer.
Author Response
I really appreciate your comments.
I revised manuscript as you mentioned and added authors’ opinion regarding the age-matched study as below.
Comment 1: Authors should mention that they were not able to match age between cases and controls due to design of this study
Reply to reviewer:
I am thankful to your comments. As your opinion, age-matched study is requested sometimes. In this study, it wasn't impossible to do it, but didn't want to. We enrolled the patients who were supposed to be taken the surgery for pelvic masses in order, as described in the reply of the 1st revision. We needed the natural patients group for each benign and cancer as unintended. These compositions of patients’ group were shown in other studies including ROMA, OVA1, and RMI.
Comment 2: Secondly, as authors replied to creatinine comment, readers should know this as a limitation of this study. They should state that due to age discrepancy between two groups, it is not clear if the significance of creatinine is affected by age or cancer.
Reply to reviewer:
I added next sentences to the end of discussion section.
“The last limitation is on the certainty of creatinine as a biomarker. The urinary creatinine concentration is changeable based on age, race/ethnicity, body mass index, and time of day. These factors were not controlled in this study. However, other biomarkers from urine change following creatinine concentration. Therefore, when using multi-biomarkers, the interaction among biomarkers is more important rather than factors associated with the creatinine level.”